# Health inequities in influenza transmission and surveillance

**Casey M. Zipfel**[1], **Vittoria Colizza**[2], **Shweta Bansal**[1]*

**1** Department of Biology, Georgetown University, Washington DC, United States of America, **2** INSERM, Sorbonne Université, Institut Pierre Louis d'Epidémiologie et de Santé Publique IPLESP, F75012 Paris, France

* shweta.bansal@georgetown.edu

## Abstract

The lower an individual's socioeconomic position, the higher their risk of poor health in low-, middle-, and high-income settings alike. As health inequities grow, it is imperative that we develop an empirically-driven mechanistic understanding of the determinants of health disparities, and capture disease burden in at-risk populations to prevent exacerbation of disparities. Past work has been limited in data or scope and has thus fallen short of generalizable insights. Here, we integrate empirical data from observational studies and large-scale healthcare data with models to characterize the dynamics and spatial heterogeneity of health disparities in an infectious disease case study: influenza. We find that variation in social and healthcare-based determinants exacerbates influenza epidemics, and that low socioeconomic status (SES) individuals disproportionately bear the burden of infection. We also identify geographical hotspots of influenza burden in low SES populations, much of which is overlooked in traditional influenza surveillance, and find that these differences are most predicted by variation in susceptibility and access to sickness absenteeism. Our results highlight that the effect of overlapping factors is synergistic and that reducing this intersectionality can significantly reduce inequities. Additionally, health disparities are expressed geographically, and targeting public health efforts spatially may be an efficient use of resources to abate inequities. The association between health and socioeconomic prosperity has a long history in the epidemiological literature; addressing health inequities in respiratory-transmitted infectious disease burden is an important step towards social justice in public health, and ignoring them promises to pose a serious threat.

## Author summary

Health inequities, or increased morbidity and mortality due to social factors, have been demonstrated for respiratory-transmitted infectious diseases, most recently highlighted by disparities in COVID-19 severe cases and deaths. Many potential causes of these inequities have been proposed, but they have not been compared, and we do not understand their population-scale impacts. Our understanding of these issues is further hindered by epidemiological surveillance, which has been shown to overlook areas of low socioeconomic

**Data Availability Statement:** All data associated with this study are provided in a GitHub repository: https://github.com/bansallab/fluSES.

**Funding:** Research reported in this publication was supported by the National Institute Of General Medical Sciences of the National Institutes of

Health under Award Number R01GM123007 (SB, https://www.nih.gov/). The content is solely the responsibility of the authors and does not necessarily represent the official views of the National Institutes of Health. We also acknowledge support from the PhRMA Foundation, the Chateaubriand Fellowship Program, and the Georgetown Global Health Initiative (CMZ, https://www.phrma.org/en, https://www.chateaubriand-fellowship.org/, https://globalhealth.georgetown.edu/). The funders had no role in study design, data collection and analysis, decision to publish, or preparation of the manuscript.

**Competing interests:** The authors have declared that no competing interests exist.

status. Here, we combine mechanistic and statistical modeling with high volume datasets to disentangle the drivers of respiratory-transmitted disease disparities, and to estimate locations where these health inequities are most severe, using influenza as a case study. We show that low socioeconomic status individuals disproportionately bear the burden of influenza infection, and that all proposed factors are synergistic in their effect. Additionally we identify geographical hotspots of poor disease surveillance among populations of low socioeconomic status, which contribute to an underestimation of health disparities. As the divide in health inequities, driven by income inequality and systemic racism, grows wider across the United States, we highlight the need to understand the mechanisms that may be at the root of disparities, and we advocate for the prioritization of capabilities to monitor outbreaks in at-risk populations so that we may prevent exacerbation of inequities.

## Introduction

Health disparities are differences in health outcomes between social groups, and they persist in all modern public health settings. Health disparities may be the result of health inequalities, which are caused by biological or cultural variations, or by health inequities, which are driven by unfair factors and are avoidable with policy action [1]. There is extensive evidence that social factors, including education, employment, income, race, and ethnicity have a distinct influence on how healthy a person is: the lower an individual's socioeconomic position, the higher their risk of poor health for both chronic and infectious diseases in low-, middle-, and high-income settings alike [2]. There is also a role played by geographic context: the spatial distribution of disparity in health cannot be explained by variation in social factors alone [3]. As the divide in health disparities grows wider across the world and within countries, it is imperative that we continue to understand how social determinants impact health, and how this is reflected geographically [4]. Here, we integrate empirical insights from past studies to characterize the impact of social determinants on the dynamics and spatial heterogeneity of an infectious disease case study, influenza.

Influenza is a respiratory-transmitted infectious disease that occurs in annual epidemics in temperate regions that can have severe outcomes, especially in young children and elderly individuals [5]. Several studies have demonstrated social differences in influenza morbidity and mortality [6–11]. The most impoverished areas have been shown to experience twice the influenza hospitalizations compared to regions with the lowest rates of poverty [12], and low education has been shown to be positively associated with influenza hospitalization rates [13]. Past work has even shown that socioeconomic factors played a significant role in the morbidity and mortality caused by the 1918 influenza pandemic [14–16]. The proposed determinants of disparities in influenza burden include a number of socio-behavioral and healthcare-based dimensions [17, 18]. In particular, influenza vaccine coverage and healthcare access are higher in areas with increased levels of education and household income [19, 20]. Additionally, low socioeconomic status (SES) individuals have been shown to experience increased susceptibility to respiratory infections due to increased stress [21, 22] and have less access to paid sick leave, resulting in less school and workplace sickness absenteeism, defined as remaining home due to illness [23, 24]. Lastly, it has been proposed that the social patterns of low SES populations affect their influenza risk: larger household sizes and higher population density may lead to higher infection risk [25, 26], while a less robust social network might result in decreased exposure, but also less support during recovery if infected [18].

Mathematical modeling studies of social disparities in influenza burden have used a simulation approach [27–29] and have focused on the effects of material deprivation (i.e. lack of access from income, education, and employment) or social deprivation (i.e. lack of social cohesion and support due to small household sizes, single parenting, divorce or widowing). Such studies are important in uncovering the mechanistic explanations of influenza disparities, but have been limited in their geographical extent, or by the use of proxy measures. For example, [27, 29] consider phenomenological variation in social contact rates without empirical evidence linking vulnerable groups to that variation, thus limiting insights on the mechanisms that lead to influenza disparities; [28, 29] focus on dynamics within specific cities, limiting generalizability.

Surveillance-based statistical studies of influenza disparities have been spatial in nature and have highlighted the challenges of disease surveillance under these disparities. Surveillance systems gather the data that shapes our understanding of influenza dynamics, and in the US and most European countries, influenza-like illness (ILI) surveillance occurs through reporting by sentinel healthcare providers. Such sentinel surveillance systems have been resource-efficient means of collecting high quality data, but they do not reliably capture data for all populations, since they are dependent on health care accessibility, health care seeking behavior, and other reporting issues [30, 31]. As a result, studies that rely on healthcare data for characterizing rates of ILI sometimes find decreasing rates of disease with increasing social deprivation [18]. While this negative association may be the result of lower exposure in impoverished areas (as suggested by [18]), it is likely that there exist spatial and social heterogeneities in surveillance caused by healthcare utilization. Indeed, Scarpino et al. have shown that the most impoverished areas are blindspots in the US influenza sentinel surveillance system, ILINet, and models based on these data make the best predictions in affluent areas, while making the worst predictions in impoverished locations [32]. To better understand and respond to influenza epidemics and pandemics, we must improve our capability to detect and monitor outbreaks in at-risk populations.

In this work, we (a) develop data-driven epidemiological models to assess how social and healthcare-based determinants impact population-level influenza transmission in a controlled manner; and (b) develop statistical ecological models from large-scale disease data to estimate latent influenza burden in vulnerable populations in the United States. We hypothesize that low SES populations bear a disproportionate burden of influenza infection, and that a combination of social, economic and health factors cause this disparity. We aim to identify geographic areas where burden is highest in low SES populations to provide hotspots for additional surveillance. As health disparities widen, it is imperative that we develop an empirically-driven mechanistic understanding of the determinants of health disparities, and capture disease burden in at-risk populations. Such insights can allow for improved influenza forecasting, resource allocation and targeted intervention design.

## Results

Here, we have evaluated the impact of social and healthcare-based mechanisms on driving influenza disparities. We achieved this through three main steps. First, we used population network modeling to generate contact networks that represent realistic social contact based on SES and modeled influenza transmission on these realistic contact networks, incorporating hypothesized drivers of influenza inequities. This framework better allowed us to understand the role that SES-driven variation plays in determining influenza dynamics and allowed us to disentangle the effects of multiple proposed drivers of influenza transmission among those of differing SES. Second, we ecologically investigated the impacts of low SES on influenza. We

estimated low SES ILI incidence ratios at the county-level in half of the states in the US using a spatial inferential model, accounting for transmission trends identified in the prior epidemiological model experiments, variation in social, economic and health factors, and measurement biases. This model allowed us to identify areas which may be currently overlooked by influenza surveillance systems. These findings also indicate potential SES-based factors associated with disproportionate burden at the population level, which could guide future public health efforts to reduce socioeconomic health disparities.

## Contact patterns vary by socioeconomic status

Contact patterns have been demonstrated to vary by socioeconomic status [18], but we have lacked social contact networks that explicitly incorporate these differences. To enable testing of hypotheses about social contact trends, we used an egocentric exponential random graph model (ERGM) to simulate networks with realistic social contact patterns based on socioeconomic status (measured by education level, [33]) from the POLYMOD social contact survey, a large social contact survey conducted across Europe [34]. The fitted network model is consistent with the contact heterogeneity in the data (Fig 1A), and all individual-level attributes (i.e. age, sex, contact location, and education level) are significant in predicting contact structure (S1 Table). Additionally, we incorporated varying levels of low SES individuals into the networks to investigate hypotheses in populations with varying SES composition, from 20% low SES to 60% low SES. The resulting networks are consistent in network structure based on degree and assortative degree (number of contacts with those of the same attribute/number in network with that attribute) by SES-status (Fig 1B). Thus, networks with increased representation of low SES individuals maintain the same SES-based contact patterns as the POLYMOD data. Importantly, the network model captures variation in contact structure by SES. In particular, low SES individuals have lower mean degree and variation in degree (Fig 1C), but have higher SES-assortative degree compared to those of higher SES (Fig 1D).

## Inequities increase low SES influenza transmission

There appears to be variation in contact trends dependent on socioeconomic status, thus it is important to consider how this network structure impacts epidemiological dynamics. To assess the role of social and healthcare-based heterogeneity, we integrated into an epidemiological network model of influenza transmission five key hypothesized drivers of disparities in influenza burden: a) social contact differences, or fewer social contacts and higher assortativity (as represented in our empirically-informed contact network model); b) low vaccine uptake; c) low healthcare utilization, which results in less access to influenza antivirals; d) high susceptibility, which results from stressful environmental factors; and e) low sickness absenteeism from school or work. S4 Table explains how these drivers are incorporated into the influenza transmission simulation, and includes other epidemiological parameters. The "low" parameters apply to low SES individuals, and "high" parameters apply to medium and high SES individuals. Then, we simulate influenza with these parameters randomly distributed across the population at the same rate as a positive control. Fig 2A shows the infection burden of low SES individuals (i.e. the ratio of the number low SES infections and the number of all infections) in the presence of each factor, combined with social cohesion (included in the network structure). Each factor results in a significant increase in the low SES infection burden in the presence of SES-based variation in parameters (dark green) compared the the random distribution of parameters (light green), and the effect is most pronounced when all the factors occur simultaneously. In contrast, the epidemic size (i.e. the ratio of the number of infections and the population size) for the positive control is larger than the SES-heterogeneous treatments,

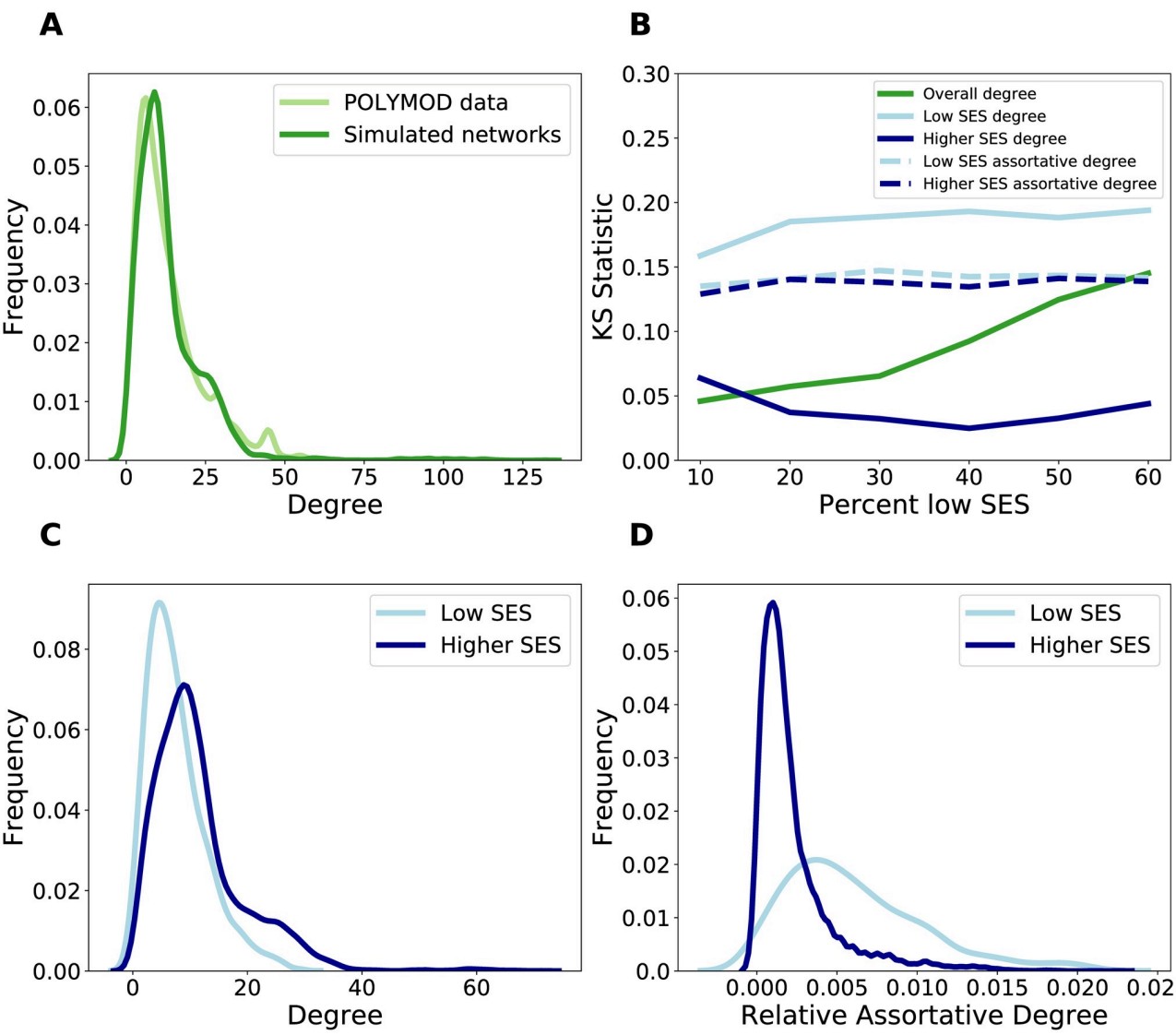

**Fig 1. The characteristics of the networks generated from the ERGM model based on POLYMOD data.** A: The degree distribution of the POLYMOD data (light green) compared to 10 simulated networks (dark green). B: The Kolmogorov-Smirnov (KS) statistic to evaluate the dissimilarity of the ERGM-simulated networks to the POLYMOD data as additional low education individuals are added to the network. KS statistics compare the dissimilarity of the overall degree distribution (dark green), the degree distribution of low SES nodes (light blue, solid), the degree distribution of high SES nodes (dark blue solid), the assortative degree (e.g. the low SES contacts of low SES nodes) for low SES nodes (light blue, dashed), and the assortative degree for high SES nodes (dark blue, dashed). Low KS values indicate similar distributions. C: The degree distribution of low SES nodes (light blue) and high SES nodes (dark blue) in 10 simulated networks. D: The relative assortative degree distribution (e.g. number of low SES contacts of low SES nodes/number of low SES nodes) of low SES nodes (light blue) and high SES nodes (dark blue) in 10 simulated networks.

for all treatments (with the exception of the increased stress treatment) (S28 Fig). A sensitivity analysis of the low SES parameter values demonstrates that our findings of increased infection of low SES individuals when they experience transmission-increasing mechanisms, compared to when the transmission-increasing mechanisms are randomly distributed, hold for all considered values (S31–S34 Figs). Thus, wherever there is a disparity in these parameters, our findings hold, regardless of the magnitude of the disparity.

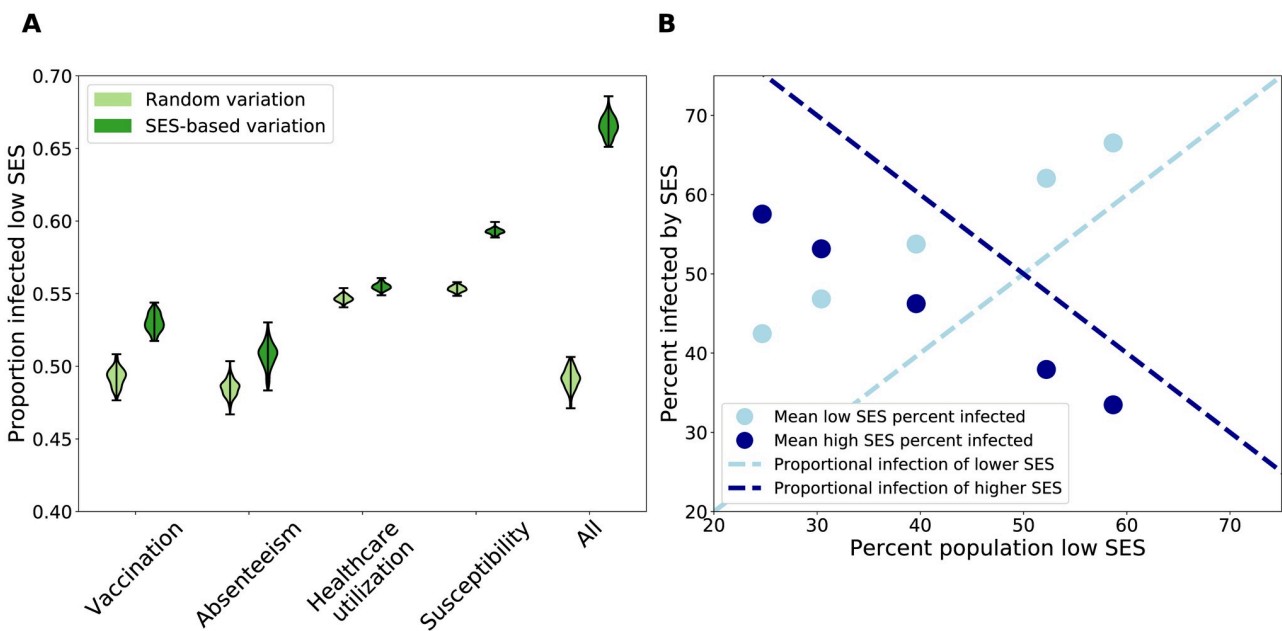

**Fig 2. Results of epidemiological simulations on ERGM networks with SES-driven social and healthcare-based differences.** A) All of the proposed SES-driven differences result in an increase in infection of low SES individuals (dark green, right of paired violin plots), compared to simulations where the differences are randomly distributed throughout the population (light green, left of paired violin plots). This difference is most pronounced when all of the mechanisms occur together. These simulations were performed on a network composed of 60% low SES, but the results are consistent across networks with different SES compositions. B) In all networks, when all SES-driven differences are present, low SES individuals (mean percent of infected population that is low SES shown in light blue dots) are disproportionately infected, relative to the expectation (light blue dashed line). High SES individuals are disproportionately underinfected compared the expectation (dark blue dots compared to dark blue dashed line).

This combination of results can be explained by the role that low SES individuals play in the network. On the one hand, low SES individuals have lower mean degree (Fig 1C). When these low degree individuals experience transmission-increasing mechanisms, this results in a smaller epidemic size, compared to the scenario where high SES, and high degree, individuals experience the same mechanisms. Thus, when SES-driven processes that increase transmission affect low SES individuals, it results in a smaller overall epidemic. On the other hand, low SES individuals have high assortativity with other low SES individuals (Fig 1D). Thus, when health disparities increase transmission for low SES individuals, they are more likely to infect other low SES individuals that are also experiencing these mechanisms, resulting in increased spread among this assortative group. This result highlights the need for surveillance and research focused on low SES populations, as the emergent high infection burden of low SES, at-risk individuals could be overlooked due to lower epidemic sizes when aggregated.

Next, we consider how low SES infection burden scales with an increasingly large low SES population. We find that epidemic size increases with an increasing proportion of low SES individuals, and this effect appears to be driven by increasing infection of low SES individuals as they make up a larger component of the network (S27 Fig). Indeed, low SES individuals experience a disproportionately large infection burden when all SES-based factors occur (Fig 2B). Additionally, high SES individuals experience a disproportionately small infection burden in the presence of the same factors. This distribution of infection also is consistent over multiple influenza seasons, with partial polarized immunity to reinfection. Though the overall epidemic size changes due to prior immunity (S29 Fig), low SES individuals are dispro-portionately infected in each season (S30 Fig).

### Low SES infection burden is spatially heterogeneous, and high in the southeastern US

Our results thus far characterize the mechanistic role that social and healthcare-based factors play on influenza burden in low SES populations in data-driven controlled experiments. Here, we aim to characterize how macroscopic factors impact influenza dynamics in low-SES populations, integrating our theoretical findings with population-level data. For population-level influenza data, we used medical claims of ILI at the county level in 25 states in the US, based on sufficient data availability. This data stream has been demonstrated to provide enhanced surveillance opportunities for influenza-like illness [31, 35]. However, when we compare county ILI burden with the proportion of the county's population of low SES, we find a negative relationship, indicating lower levels of influenza in counties with larger low SES populations (S35(A) Fig). This pattern is counter to our previous mechanistic model findings and to past small scale studies, suggesting that there may be measurement biases in these surveillance data (S35(B) Fig).

To better estimate influenza burden in low SES populations, we fit a Bayesian spatial hierarchical model that accounts for measurement biases and borrows information from spatial covariates pertaining to low SES individuals and the mechanistic modeling experiments. Here, we define low SES ILI as an incidence ratio, where low SES ILI cases are normalized by the number of 1000 visits, to account for spatio-temporal variation in database coverage and healthcare-seeking. Thus, the model outcome data is the rate of ILI healthcare visits per 1000 healthcare visits within each county. Our model estimates of this low SES ILI incidence ratio show a positive relationship with low SES population size (S35(C) Fig), and allow us to consider spatial disparities in influenza burden. Fig 3 shows the county-level map of the low SES ILI incidence ratio. This map highlights areas with a high incidence ratio among low SES individuals in the southeastern United States, which is a region where low socioeconomic status is prevalent. This also demonstrates that there are significant levels of heterogeneity both within

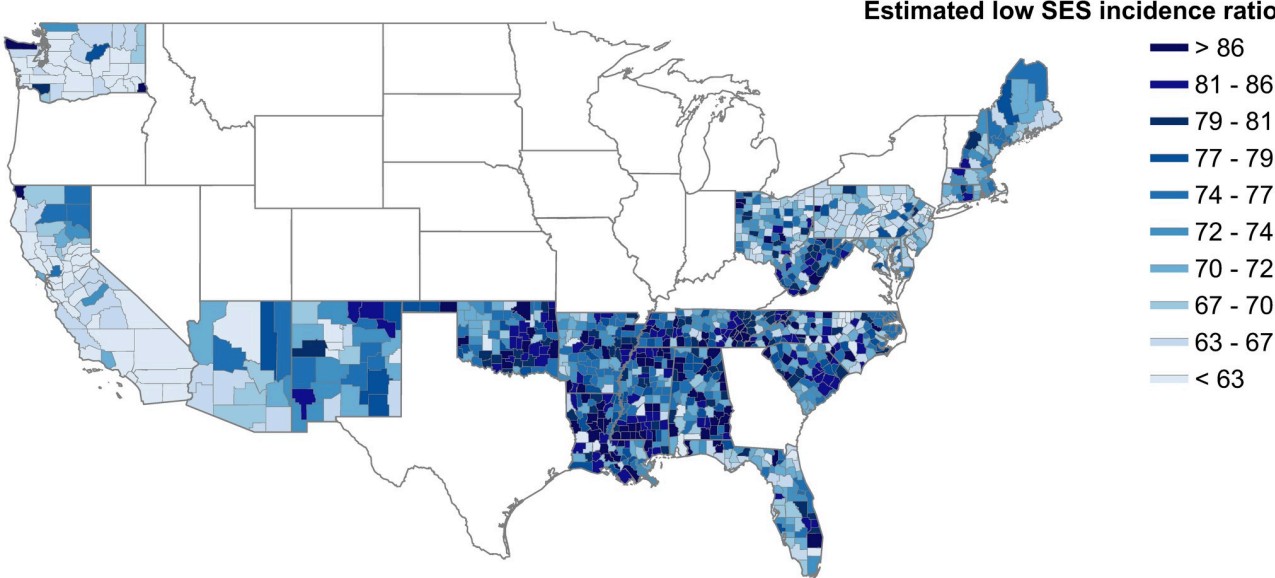

**Fig 3. County-level map of model estimates of low SES ILI incidence ratio per 1,000 people.** Lower values are represented in light blue, and higher values are represented in darker blue. States in white were omitted due to lack of covariate data. Some county covariate data in included states was imputed based on surrounding neighbors, where missing. Unimputed findings are available in S37 Fig.

and between states. These estimates can guide targeting of improved surveillance and steps to alleviate the influenza burden in low SES populations.

To validate our findings, we grouped our model estimates by county-level poverty rates, and compared the incidence ratio to prior population-level studies that correlate influenza rates and poverty levels, though these studies do not focus on low SES individuals, so the comparison is not direct. We find increasing low SES ILI incidence in areas with increasing levels of poverty, which agrees with trends in [11, 12] (S40 Fig).

## Susceptibility and sickness absenteeism differences may be associated with ILI in low SES populations

Fig 4 shows the coefficient estimates and credible intervals resulting from the Bayesian spatial hierarchical model. Levels of poor health among low SES individuals, as a measure of susceptibility to infection, are positively related with low SES ILI incidence. Thus, areas with higher reports of poor health among low SES individuals are associated with higher burden of ILI among low SES populations. Also, access to sickness absenteeism among low SES individuals, represented by the number of low SES students that are absent for more than 10 days in a school year, is negatively related to low SES ILI incidence ratios. Thus, areas where more low SES students are able to be absent are associated with lower rates of low SES ILI.

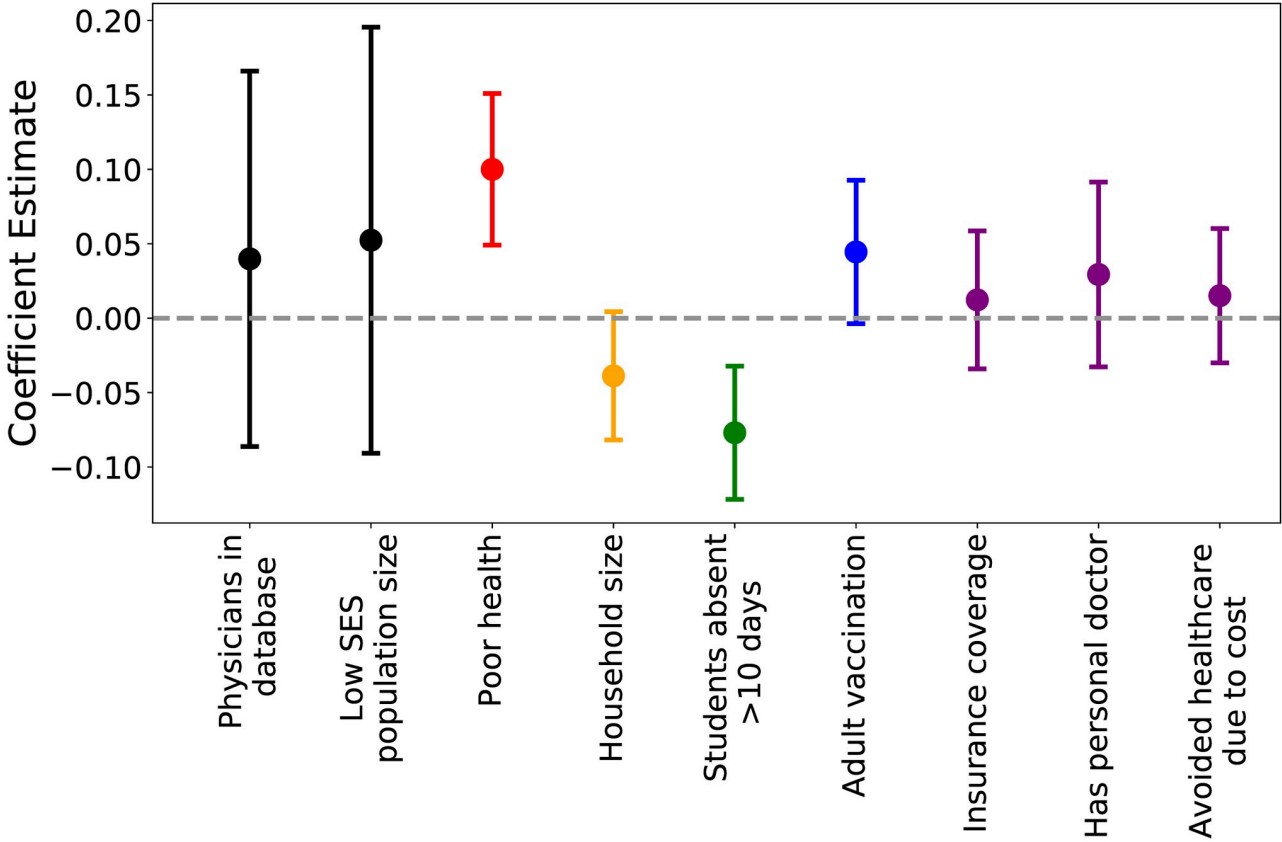

**Fig 4. Mean model coefficient estimates and credible intervals.** Points are colored by what process each covariate represents (black: measurement bias, red: susceptibility, orange: social contact differences, green: sickness absenteeism, blue: vaccination, purple: healthcare utilization). Each process covariate is specific to low SES populations (e.g."adult vaccination" is only vaccination rates of low SES adults).

## Discussion

Increased infectious disease prevalence among lower socioeconomic status populations has been observed in many settings. What has been missing, however, is a better understanding of the mechanisms that drive this disparity. We used a mechanistic epidemiological network model which allowed us to assess the impacts of SES-based social and healthcare-based differences on influenza in controlled experiments. This highlighted the role played by all mechanisms in tandem to produce disproportionate disease burden in low SES populations. To address the gap that exists in our surveillance of ILI and to estimate the spatial distribution of influenza disparities, we then used a Bayesian spatial hierarchical model to estimate population-level low SES ILI at a fine spatial scale across the United States, accounting for disproportionate infection of low SES individuals, measurement biases, and county-level factors hypothesized to be associated with influenza and SES. Our results shine light on the spatial distribution of respiratory-transmitted disease health disparities.

In our epidemiological model, disease transmission occurs over the contact network structure, which accounts for heterogeneity in contact patterns by SES. While past work has integrated contact heterogeneity by other socio-demographic characteristics such as age and occupation [36, 37], SES-based contact heterogeneity has not been integrated into contact network models for epidemiological purposes. Epidemiological simulations on the SES-heterogeneous network reveals that each hypothesized factor leads to increased infection of low SES individuals. Additionally, we find that communities with larger low SES populations experience larger epidemics, which is in agreement with prior studies [10–12]. The proposed drivers are not mutually exclusive, so this reveals potential effects that could not be identified in past studies that investigate the impact of a single SES-based mechanism or impacts that might be aggregated in observational studies. We note that these experiments also include SES-based variation in social cohesion (i.e. SES-based contact heterogeneity in the population model), so the effect shown in Fig 2A is the result of both mechanisms combined. In S28 Fig, we also illustrate the impact of each mechanism independent of social cohesion.

Our efforts to consider the impacts of low SES on influenza spatial heterogeneity generated county-level maps of ILI incidence in low SES populations. Our findings identify pockets of high ILI burden in low SES populations across the United States, and represent a first step in filling the gap that exists in all healthcare-based surveillance. The model also produced a set of estimates for the effect of each hypothesized ecological measure. We find that low sickness absenteeism and high susceptibility are significantly associated with influenza in low SES populations. This supports our previous finding that multiple mechanisms compound to result in disproportionate low SES influenza burden. Low SES absenteeism is here measured by student absenteeism, which may not be a perfect measure of sickness absenteeism or paid sick leave access. However, other fine-scale data was lacking, and a student's ability to be absent is related to a parent's ability to be home to care for the child, and differences in access to paid sick leave by SES have been related to student sickness absenteeism levels due to influenza [38–41]. To validate our findings, we compared our model estimates to previous estimates of influenza incidence ratios, stratified by poverty level. This is not a direct comparison, as previous studies present the incidence ratios for the entire population, not just for low SES individuals within those populations. Our results show more consistently high incidence ratios compared to the larger increases between poverty rates in prior studies. We attribute this to the incorporation of the measurement process into our models, which accounts for undersurveillance of low SES infection, whereas healthcare access and healthcare seeking differences may have missed low SES cases in prior studies. Ideally, data on respiratory infection of low SES individuals would be available at a fine spatial scale to more directly assess the validity of our models, but the lack

of such a dataset highlights the need for future surveillance and data collection that focuses attention on lower SES populations.

Our work has several limitations. The network structure of our epidemiological model is based on one social survey from 2007 in Europe, and may be less representative of the United States today. Additionally, survey data was not collected for the SES of the contacts of survey participants, which required us to make assumptions which could affect our results about SES assortativity. Additional social contact data collection across the United States that accounts for SES heterogeneity would be useful for future studies given the large socioeconomic inequality in the country [42, 43]. In our spatial ecological model, we assume that disproportionate burden in low SES populations remains constant over influenza seasons. While there may be variation in the dynamics of ILI among low SES populations over time, time-varying data on our covariates is currently unavailable and we do expect socioeconomic factors and health behaviors to remain relatively consistent across seasons. Future work could focus on temporal variation in low SES ILI dynamics. Additionally, our spatial ecological model is only able to provide estimates for half of the states in the US, and the states are mostly on the coasts. This highlights the need for more data collection pertaining to low SES individuals, not only for epidemiological data, but also for a wide variety of other topics to provide covariate data and to create a better understanding of at-risk populations.

A main limitation of the spatial inferential model is the identifiability of separate effects. Here, we have identified possible associations in our model, but this is only the start to disentangling the factors that contribute to health inequities. The lack of significant association with the other incorporated covariates does not indicate that these are not important to inequities in influenza transmission in low SES populations. These impacts may be obscured by several issues. The covariate data may be impacted by its own biases, insufficient sample sizes, and other limitations. When ubiquitous systemic inequities go unaccounted for in data collection and processing, the signal of low SES individuals may be obscured. We aimed to counteract this by only using covariate data specific to low SES populations, but this was parsed out from data collected for the whole population that included demographic data, identifying potentially lower SES individuals. Next, there may be other factors relating to increased influenza transmission that may not be identifiable when focused on mechanistic explanations, and the model may not be able to parse synergistic factors. Our network epidemic model demonstrates that multiple factors of inequity can compound one another non-linearly, and statistically identifying individual effects remains a challenge due to lack of data and statistical limitations. Further attention to systemic inequities in health and epidemiology will be necessary to move this problem forward.

As the divide in health inequities, driven by income inequality and systemic racism, grows wider across the United States, we propose the use of infectious disease case studies to improve our understanding of this challenging problem. We suggest that we move beyond studies based on proxy measures such as income and education which may provide an incomplete picture, and dig into the mechanisms that may be at the root of inequities. Furthermore, we advocate for the prioritization of capabilities to detect and monitor outbreaks in at-risk populations so that we may prevent exacerbation of inequities. Addressing health inequities in respiratory infectious disease burden is an important step towards social justice in public health, and ignoring them promises to pose a serious threat to the entire population. Indeed, the damaging impacts of health inequities for respiratory infectious diseases have already been highlighted in the COVID-19 pandemic [44]. Our results suggest that (a) the effect of overlapping behavioral and social factors is synergistic and reducing this intersectionality can significantly reduce inequities; and (b) health disparities are expressed geographically and targeting public health efforts spatially may be an efficient use of resources to abate inequities. Further

attention to the mechanisms and processes that lead to health inequities, and specifically health inequities that may be overlooked by our currently surveillance systems, will be important to identifying actionable steps to mitigate negative health outcomes in the future.

## Materials and methods

In this study, we use (1) a mechanistic network epidemiological model to assess influenza transmission in the presence of individual-level socioeconomic status (SES)-based social and healthcare-based variation; and (2) an inferential spatial model to geographically localize influenza-like illness (ILI) burden among low SES populations in the presence of population-level variation in social and health indicators. Data and code for the implementation of these methods are available at [45].

### Modeling the impact of individual-based SES factors on disease burden

To achieve the mechanistic understanding, we (a) fitted a contact network model from empirical contact data that includes contact heterogeneity stratified by age, sex, contact location, and socioeconomic status; and (b) performed epidemiological simulations on these networked populations integrating epidemiological differences based on SES, parameterized by empirical studies.

**Contact network model.**   In a contact network model, nodes represent individuals, and edges represent epidemiologically-relevant interactions between individuals. The degree of a node is the number of edges, or contacts, of the node, and the degree distribution of a network is the frequency distribution of node degrees within the population. To generate realistic contact networks to evaluate epidemic outcomes, we used an egocentric exponential random graph model (ERGM) [46]. An egocentric ERGM allows for the construction of sociocentric networks based on egocentrically sampled data, in which participants (or *egos*) report the identity of their contacts (or *alters*), who may or may not be study participants. Our egocentric ERGM model was based on the POLYMOD dataset, a large, egocentric contact survey that identifies close interactions of over 7000 individuals across eight European countries [34].

Nodes in the network had the following attributes: (a) age, grouped as infants-toddlers (age 0-4), school-aged children (age 5-18), adults (age 19-64), and elderly (age 65-100); (b) sex, classified as male or female; (c) contact location, in which a node can have known home contacts and known school or work contacts; (d) education level as a proxy for socioeconomic status [33], grouped as low education (less than a high school education), medium education (high school or vocational school education), or high education (any university education or beyond). Age and sex were available in the data for egos and alters, while education level was only provided for egos. Therefore, it was assumed that an ego's work contacts had the same education level based on their occupation, and that an ego's home contacts had the same education level as an indicator of household socioeconomic status. To represent communities with different SES compositions, we resampled additional low education egos from the low education sample in the POLYMOD dataset. These networks allow us to examine how epidemic dynamics might differ in populations with different proportions of low SES individuals in the population (e.g. capturing the SES variability observed in the United States). We produce networks composed of approximately 20-60% low education individuals (S3 Table).

The model was fit using the ERGM package [47, 48]. The best model was selected based on multicollinearity criteria and goodness of fit to the POLYMOD data. The best model observed data was the egos and their alter contacts. Model terms included edges, node attributes for sex, age, school/work, and education, and homophily for age, home, school/work, and education. From the best fit ERGM model, we simulated 10 networks. Additional model details (S1 Appendix), model terms (S1 Table), multicollinearity measures (S2 Table), model diagnostics,

and goodness of fit measures (S1–S26 Figs) are available. We highlight that we have selected the best fit model, balancing the fit of each incorporated model term. Thus, while each individual attribute may not be an exact fit, this model best captures the main characteristics of each attribute.

Random regular networks of the same size and mean degree were also generated as null networks to evaluate the effect of contact heterogeneity. We used the Networkx package for network generation and analysis [49].

**SES-based epidemiological model.** Chain binomial SEIR (Susceptible-Exposed-Infected-Recovered) simulations were performed on the networks generated by the egocentric ERGM model and the random control networks to examine the spread of a respiratory infection, like influenza, through a naive population. Model parameters pertinent to seasonal influenza spread were selected from literature (S4 Table) [50, 51].

Five hypothesized drivers for increased influenza in low SES populations were integrated into the epidemiological simulations. Each hypothesized driver represents a social or health-based factor. Each hypothesized driver has 2 relevant parameters: one that pertains to high SES individuals and one that pertains to low SES individuals. These values were selected from literature (S4 Table). We conducted a sensitivity analysis of the robustness of our findings to different low SES parameters (S31–S34 Figs). The first hypothesized driver of influenza transmission inequities is social contact differences, which represents the SES-based social contact rates of individuals, and thus is represented by the ERGM-generated networks. The remaining factors are:

- Low vaccine uptake: Individuals may be vaccinated before the start of the season with a perfectly efficacious vaccine. Vaccinated nodes were randomly selected and removed from the network. Vaccination coverage is parameterized by $\delta_{high}$ and $\delta_{low}$ in high- and low-SES individuals, respectively. The value of delta was based on a US population survey of vaccine coverage related to education level [19].

- High susceptibility: Those who experience a more stressful environment are more susceptible to infection, and thus have a greater probability of becoming infected upon contact with an infected individual. Susceptibility is parameterized by $\beta_{high}$ and $\beta_{low}$ in high- and low-SES individuals, respectively. This is based on an immune challenge experiment that found that those of high SES were about half as likely to become infected with a cold compared to those of low SES [22].

- Low healthcare utilization: Infected individuals who do not seek healthcare and receive antivirals have a longer infectious period, based on a model of within-host and population-level dynamics [52]. The proportion of the infected population seeking healthcare is parameterized by $\gamma_{high}$ and $\gamma_{low}$ in high- and low-SES individuals, respectively.

- Low sickness absenteeism: Infected individuals may exhibit sickness absenteeism from school or work if they have access to leave and care at home. Those exhibiting sickness absenteeism remove 90% of contacts [53]. Access to sickness absenteeism is parameterized by $\rho_{high}$ and $\rho_{low}$ in high- and low-SES individuals, respectively. These values are based on rates of paid sick leave by education level in a survey across the US [54].

For our experimental design, each SES-based factor was tested separately and together on each network. The high parameters were applied to medium and high education nodes, and the low parameters were applied to low education nodes, as defined in the ERGM model above. Disease outbreaks for each treatment were simulated 200 times on each network, with 5 replicate networks. We assume a naive population of entirely susceptible individuals, and each

simulation represented one influenza season, continuing until there were no new exposed individuals. We note that the only isolation of infected individuals that occurs is when absenteeism is incorporated into the simulation, as described above. We also considered two controls to compare our experimental results: a) a homogeneous control, in which the high and low parameters were randomly distributed across a random regular network; b) a heterogeneous control, in which the high and low parameters were randomly distributed across the ERGM-generated networks. We ensured that the number of individuals treated based on each parameter remained constant across the simulation types: random parameter distribution on regular network, random parameter distribution on ERGM-generated network, and SES-based parameter distribution on ERGM-generated network.

We also investigated whether our findings were maintained across multiple influenza seasons, where immunity in prior seasons would alter viral transmission on the network structure. The aim of this experiment is to assess the relationship between pre-existing immunity and low SES individuals, and we used a simplified approach to model loss of immunity across seasons [55] and measured the disproportionate burden of disease among low SES populations (more details in S2 Appendix).

### Modelling surveillance of disease in low SES populations

To achieve an inferential understanding, we (a) integrated the network model findings with empirical ILI data for an estimate of ILI burden among low SES individuals; and (b) fitted a spatial Bayesian hierarchical model with population-level covariates to account for measurement biases and improve our estimate of low SES ILI burden at the population-level.

**Spatial inferential model.** We used a Bayesian spatial hierarchical model to estimate latent ILI cases among low SES individuals. This is an N-mixture model, which accounts for imperfect detection of low SES ILI cases through a measurement process, as well as borrowing information from county-level factors associated with influenza in low SES populations. The goals of this model are to estimate cases of ILI in low SES populations in counties across the United States accounting for measurement processes and data on SES-based social and healthcare differences, and to identify the relationship between the hypothesized drivers of inequities and low SES ILI at the population level. We modeled low SES ILI ($Y_{it}$) in county $i$ in flu season $t$ as:

$$Y_{it}|N_i \sim Binomial(N_i, p_{i,t})$$

where $p_{i,t}$ is the probability of detecting low SES ILI cases, and $N_i$ is the true ILI cases among low SES individuals.

We modeled the probability of detection $p_{i,t}$ as:

$$logit(p_{i,t}) = \alpha_0 + \sum_1^k \alpha_k z_{i,t,k} + v_c + v_s$$

where $\alpha_0$ is the intercept, $\alpha_k$ represents the coefficient estimate for the $k$ measurement process predictor variables, $z_{i,t,k}$ (here, physicians in database and low SES population size), and $v_c$ and $v_s$ are group effects for county and state, respectively.

We modeled the latent low SES ILI cases as:

$$N_i \sim NegBin(\lambda_i, \theta)$$

where the negative binomial distribution is parameterized by probability $\lambda_i$ and size $\theta$.

The $\lambda_i$ is modeled by:

$$log(\lambda_i) = \beta_0 + \sum_1^j \beta_j x_{i,j} + \mu_c + \mu_s$$

where $\beta_0$ is the intercept, $\beta_j$ represents coefficient estimates for the $j$ low SES ILI process covariates, $x_{i,j}$ (here, variables that capture each of the hypothesized mechanisms- susceptibility, social contact differences, absenteeism, vaccination, and healthcare access- in low SES populations), and $\mu_c$ and $\mu_s$ represent county-level and state-level group effects, respectively. We performed approximate Bayesian inference using Integrated Nested Laplace Approximations (INLA) with the R-INLA package [56]. INLA has demonstrated computational efficiency for latent Gaussian models, produced similar estimates for fixed parameters as established implementations of Markov Chain Monte Carlo (MCMC) methods for Bayesian inference, and been applied to disease mapping and spatial ecology questions. We evaluated DIC, WAIC, model residuals and compared modeled and observed outcomes in order to assess model fit. Additional model details can be found in S36, S38 and S39 Figs. We highlight that areas with high observed low SES ILI are underestimated by the model, due to measurement being efficient in these areas (S36 Fig). This indicates that our estimates of low SES ILI in these areas may be conservative.

**Response data.**    We define the response in our model to be the observed influenza-like illness (ILI) burden in low SES populations. In particular, we use influenza-like illness reports from a medical claims database from across the United States collected during 2002-2008. Additional details on the dataset can be found in [31, 35]. To normalize these observed counts, we take the ratio of ILI visits for every 1000 visits for any diagnosis during the influenza season. We refer to this value as an incidence ratio. These data are at the county-level but are not stratified by SES. To produce a county-level estimate of ILI in low SES populations for our spatial model, we use the observed ILI burden in the total population and scale this by the proportion expected among low SES individuals as predicted by the epidemiological model from the first part of our study (as summarized in Fig 2B). As an example, a hypothetical county composed of 40% low SES individuals has 500 total ILI visits out of 8,000 total healthcare visits. Fig 2B shows that a county with 40% low SES in the population is expected to have about 55% of ILI cases in low SES individuals. Thus, we estimate this county's low SES ILI cases as 275 ILI cases, which we normalize per 1000 total visits, resulting in a rounded ILI incidence ratio of 34.

**Covariate data.**    All covariate data are at the county level, and are centered and standardized. We make the assumption that county characteristics remain relatively constant over time, and harness together covariate data from different years based on availability and coverage, and make the assumption that factors remain relatively constant from 2002-2008. All covariate data was evaluated for multicollinearity, and all included covariates had a variance inflation factor less than 2. First, covariate data was included for the measurement submodel to characterize database coverage and population size. For database coverage, we used the number of physicians reporting to the medical claims database, which was reported by the database and averaged over reported years. Additionally, the population of low SES individuals was included since that measures the size of the considered population. Low SES population size was measured as the county population size, reported by the US Census Bureau [57], multiplied by the percent of the population with less than a high school education from County Health Rankings [58]. Then, for the process model, covariate data were included as a marker for each hypothesized driver of low SES influenza. We ensured that all process covariate data pertained just to low SES populations. For a measure of susceptibility, reports of poor health in individuals with less than a high school education, divided by the sample size of low education

individuals, were collected from the the Behavioral Risk Factor Surveillance System (BRFSS) from the CDC, which is available at the individual level and reported by county in 2012 [59]. For a measure of social cohesion, mean household size reported by those with less than a high school education was also collected from BRFSS. To measure access to healthcare, rates of reporting having health insurance, reporting having a personal doctor, and reporting avoiding healthcare due to cost by those of low education were divided by low education sample size from BRFSS. To measure vaccination, reports of adult vaccination in low education individuals were divided by the low education sample size in BRFSS. To measure sickness absenteeism, the rate of chronic sickness absenteeism, or students absent for more than 10 days, was collected from the US Department of Education [60]. This data was only available stratified by race, thus the chronic sickness absenteeism reports of Black students, divided by the number of Black students, was used due to the correlation between race and socioeconomic status in the US [61]. Much of the data was available through BRFSS, which lacked coverage in many counties. Thus, counties with a low education sample size of less than 10 were omitted. Additionally, due to this sparse coverage in covariate data, we restricted our analyses to states that had complete covariate data for more than 50% of counties. This is to ensure that sparse covariate data does not skew the model, since we only want to provide inference for states that have enough data to provide reliable estimates. These challenges highlight the need for more high resolution data on low SES populations across the country. See supplement table for additional covariate data details (S5 Table).

**Imputation and validation.** Based on the assumption that counties that are close to one another are similar to one another, we imputed covariate values for missing counties in states that were included in the model. Approximately 32% of the counties in the included states had a missing covariate data value, and thus were imputed. The model was run with only the counties that had complete covariate data, thus the estimates and inference are based only on counties with complete covariate data. Then, for each missing county, we took the mean of the adjacent counties for each covariate value, to assign covariate values to the missing counties. We then used these imputed covariate values to calculate model estimates for the missing counties. The model estimates prior to imputation are available in S37 Fig. We grouped the resulting full model estimates by county-level percent living in poverty, according to Small Area Income and Poverty Estimates reported by the US Census Bureau [62]. We collected incidence/incidence ratio values reported by the same poverty level groupings from [11, 12]. Each set of incidence values was min-max normalized for comparison due to variations between reported value and population considered.

## Supporting information

**S1 Fig. MCMC diagnostics of ERGM.** MCMC diagnostics, demonstrating that appropriate MCMC sample statistics were achieved.
(EPS)

**S2 Fig. Degree distribution of male egos in POLYMOD data compared to ERGM simulated networks.** The degree distribution of the male egos in the POLYMOD data (black) and the degree distributions of males in the 10 ERGM simulated networks (gray).
(EPS)

**S3 Fig. Degree distribution of female egos in POLYMOD data compared to ERGM simulated networks.** The degree distribution of the female egos in the POLYMOD data (black) and the degree distributions of females in the 10 ERGM simulated networks (gray).
(EPS)

**S4 Fig. Degree distribution of infant/toddler egos in POLYMOD data compared to ERGM simulated networks.** The degree distribution of the infant/toddler egos in the POLYMOD data (black) and the degree distributions of infants/toddlers in the 10 ERGM simulated networks (gray).
(EPS)

**S5 Fig. Degree distribution of child egos in POLYMOD data compared to ERGM simulated networks.** The degree distribution of the child egos in the POLYMOD data (black) and the degree distributions of children in the 10 ERGM simulated networks (gray).
(EPS)

**S6 Fig. Degree distribution of adult egos in POLYMOD data compared to ERGM simulated networks.** The degree distribution of the adult egos in the POLYMOD data (black) and the degree distributions of adults in the 10 ERGM simulated networks (gray).
(EPS)

**S7 Fig. Degree distribution of elderly egos in POLYMOD data compared to ERGM simulated networks.** The degree distribution of the elderly egos in the POLYMOD data (black) and the degree distributions of elderly in the 10 ERGM simulated networks (gray).
(EPS)

**S8 Fig. Degree distribution of home egos in POLYMOD data compared to ERGM simulated networks.** The degree distribution of the home egos in the POLYMOD data (black) and the degree distributions of home nodes in the 10 ERGM simulated networks (gray).
(EPS)

**S9 Fig. Degree distribution of school egos in POLYMOD data compared to ERGM simulated networks.** The degree distribution of the school egos in the POLYMOD data (black) and the degree distributions of school nodes in the 10 ERGM simulated networks (gray).
(EPS)

**S10 Fig. Degree distribution of work egos in POLYMOD data compared to ERGM simulated networks.** The degree distribution of the work egos in the POLYMOD data (black) and the degree distributions of work nodes in the 10 ERGM simulated networks (gray).
(EPS)

**S11 Fig. Degree distribution of low education egos in POLYMOD data compared to ERGM simulated networks.** The degree distribution of the low education egos in the POLYMOD data (black) and the degree distributions of low education nodes in the 10 ERGM simulated networks (gray).
(EPS)

**S12 Fig. Degree distribution of medium education egos in POLYMOD data compared to ERGM simulated networks.** The degree distribution of the medium education egos in the POLYMOD data (black) and the degree distributions of medium education nodes in the 10 ERGM simulated networks (gray).
(EPS)

**S13 Fig. Degree distribution of high education egos in POLYMOD data compared to ERGM simulated networks.** The degree distribution of the high education egos in the POLYMOD data (black) and the degree distributions of high education nodes in the 10 ERGM simulated networks (gray).
(EPS)

**S14 Fig. Assortative degree distribution of male nodes in POLYMOD data compared to ERGM simulated networks.** The number of contacts of male egos with male alters in the POLYMOD data (black) and in the 10 ERGM simulated networks (gray).
(EPS)

**S15 Fig. Assortative degree distribution of female nodes in POLYMOD data compared to ERGM simulated networks.** The number of contacts of female egos with female alters in the POLYMOD data (black) and in the 10 ERGM simulated networks (gray).
(EPS)

**S16 Fig. Assortative degree distribution of infant/toddler nodes in POLYMOD data compared to ERGM simulated networks.** The number of contacts of infant/toddler egos with infant/toddler alters in the POLYMOD data (black) and in the 10 ERGM simulated networks (gray).
(EPS)

**S17 Fig. Assortative degree distribution of child nodes in POLYMOD data compared to ERGM simulated networks.** The number of contacts of child egos with child alters in the POLYMOD data (black) and in the 10 ERGM simulated networks (gray).
(EPS)

**S18 Fig. Assortative degree distribution of adult nodes in POLYMOD data compared to ERGM simulated networks.** The number of contacts of adult egos with adult alters in the POLYMOD data (black) and in the 10 ERGM simulated networks (gray).
(EPS)

**S19 Fig. Assortative degree distribution of elderly nodes in POLYMOD data compared to ERGM simulated networks.** The number of contacts of elderly egos with elderly alters in the POLYMOD data (black) and in the 10 ERGM simulated networks (gray).
(EPS)

**S20 Fig. Assortative degree distribution of home nodes in POLYMOD data compared to ERGM simulated networks.** The number of contacts of home egos with home alters in the POLYMOD data (black) and in the 10 ERGM simulated networks (gray).
(EPS)

**S21 Fig. Assortative degree distribution of school nodes in POLYMOD data compared to ERGM simulated networks.** The number of contacts of school egos with school alters in the POLYMOD data (black) and in the 10 ERGM simulated networks (gray).
(EPS)

**S22 Fig. Assortative degree distribution of work nodes in POLYMOD data compared to ERGM simulated networks.** The number of contacts of work egos with work alters in the POLYMOD data (black) and in the 10 ERGM simulated networks (gray).
(EPS)

**S23 Fig. Assortative degree distribution of low education nodes in POLYMOD data compared to ERGM simulated networks.** The number of contacts of low education egos with low education alters in the POLYMOD data (black) and in the 10 ERGM simulated networks (gray).
(EPS)

**S24 Fig. Assortative degree distribution of medium education nodes in POLYMOD data compared to ERGM simulated networks.** The number of contacts of medium education egos

with medium education alters in the POLYMOD data (black) and in the 10 ERGM simulated networks (gray).
(EPS)

**S25 Fig. Assortative degree distribution of high education nodes in POLYMOD data compared to ERGM simulated networks.** The number of contacts of high education egos with high education alters in the POLYMOD data (black) and in the 10 ERGM simulated networks (gray).
(EPS)

**S26 Fig. Goodness of fit of ERGM network statistics.** The black line represents the statistics of the POLYMOD data and the boxplots are the ERGM network simulated values for the statistics.
(EPS)

**S27 Fig. Epidemic size split by education level.** The mean epidemic size, or the proportion of the population infected, in influenza simulations with all SES based mechanisms occurring. The epidemic size is split by the proportion of the epidemic size composed by low SES individuals (light blue), compared to other SES individuals (dark blue). The epidemic size increases with the addition of more low SES individuals, and low SES individuals appear to make up a larger component of the epidemic size as the make up more of the population.
(EPS)

**S28 Fig. Epidemic size for each SES-based mechanism.** The impacts of each hypothesized mechanism on epidemic size. The epidemic size, or the proportion of the population that is infected, is demonstrated for each mechanism. Furthest to the left, we show the epidemic size of simulations with no mechanisms occurring. The left of the pair is the epidemic size on a regular network with the same network size and mean degree as the SES-heterogeneous network. The right of the pair is the epidemic size on the SES-heterogeneous network, simulated from the ERGM model. Next, each mechanism was randomly applied to the regular network. This is a control for network structure and SES-driven mechanisms (blue, left of sets of three boxplots). Each mechanism was also applied randomly to the SES-heterogeneous networks as a positive control, incorporating social cohesion but not SES-based differences in mechanisms (center of each set of three boxplots, light green). Lastly, each mechanism was applied to the SES-heterogeneous networks where the mechanisms impacted low SES individuals only (right of each set of three boxplots, dark green).
(EPS)

**S29 Fig. Epidemic size with polarized partial immunity.** Mean epidemic size of 5 subsequent seasons of influenza with polarized partial immunity. The number of infected individuals is split into low SES infected individuals (blue) and high SES infected individuals (orange). In each season, low SES individuals make up the majority of cases.
(EPS)

**S30 Fig. Disproportionate infection of low SES individuals with polarized partial immunity.** The proportion of the epidemic size that is composed of each SES group divided by the proportion of the network that is composed of each SES group for 5 subsequent influenza seasons with polarized partial immunity. Low SES individuals are disproportionately infected in each season (blue), and high SES individuals are disproportionately underinfected in each season (orange). The black dashed line highlights 1, which is where the bars would reach if the populations were infected proportionally to how much of the population they compose.
(EPS)

**S31 Fig. Low SES vaccination rate sensitivity analysis.** Proportion of epidemic size that is composed of low SES individuals for possible low SES vaccination rates where the low SES vaccination rate is randomly distributed (left of pair, light green), compared to where the low SES vaccination rate applies to low SES individuals (right of pair, dark green). For all values where low SES individuals are vaccinated at a lower rate than high SES individuals (black dashed line), low SES individuals are increasingly infected. The low SES vaccination rate shown in the main text is shown with the black dotted line.
(EPS)

**S32 Fig. Low SES absenteeism rate sensitivity analysis.** Proportion of epidemic size that is composed of low SES individuals for possible low SES vaccination rates where the low SES absenteeism rate is randomly distributed (left of pair, light green), compared to where the low SES absenteeism rate applies to low SES individuals (right of pair, dark green). For all values where low SES individuals are absent at a lower rate than high SES individuals (black dashed line), low SES individuals are increasingly infected. The low SES absenteeism rate shown in the main text is shown with the black dotted line.
(EPS)

**S33 Fig. Low SES gamma sensitivity analysis.** Proportion of epidemic size that is composed of low SES individuals for possible low SES gammas (representing decreased healthcare utilization and longer infectious period) where the low SES gamma is randomly distributed (left of pair, light green), compared to where the low SES gamma applies to low SES individuals (right of pair, dark green). For all values where low SES individuals recover slower than high SES individuals (black dashed line), low SES individuals are increasingly infected. The low SES gamma shown in the main text is shown with the black dotted line.
(EPS)

**S34 Fig. Low SES beta sensitivity analysis.** Proportion of epidemic size that is composed of low SES individuals for possible low SES beta (representing increased suscpetibility) where the low SES beta is randomly distributed (left of pair, light green), compared to where the low SES beta applies to low SES individuals (right of pair, dark green). For all values where low SES individuals are more susceptible than high SES individuals (black dashed line), low SES individuals are increasingly infected. The low SES beta shown in the main text is shown with the black dotted line.
(EPS)

**S35 Fig. Medical claims data ILI cases, total visits, incidence ratio observed and modeled by county low SES population.** A) ILI case counts decrease as county low SES population increases. B) Total visits in the medical claims database decrease as county low SES population increases. C) Observed and modeled low SES incidence ratio by county low SES proportion. Observed low SES ILI incidence ratio is lower and trends slightly positive. Modeled low SES incidence is high and increases more drastically as the low SES ILI population increases.
(EPS)

**S36 Fig. Model observed data versus model predicted results.** Each plot represents the data from a different influenza season, from 2002-2003 (top), through 2007-2008 (bottom). Points represent county-level data, and the one-to-one line is shown.
(EPS)

**S37 Fig. Choropleth of model estimates of incidence ratio, before imputation.**
(EPS)

**S38 Fig. Choropleth of modeled observation process, $Y_{it}$, from the Bayesian hierarchical model.**
(EPS)

**S39 Fig. Choropleth of the modeled measurement process, p, in the Bayesian hierarchical model.**
(EPS)

**S40 Fig. Min-max normalized incidence ratios related to percent in poverty.** County mean model estimates in blue, reported overall age adjusted incidence by [11] in orange, census-tract mean estimates reported by [12] in green.
(EPS)

**S1 Appendix. ERGM Model Details.** Additional details of the implementation of the Exponential Random Graph Model (ERGM).
(DOCX)

**S2 Appendix. Partial immunity sensitivity analysis Details.** Additional details on the sensitivity analysis incorporating partial immunity into epidemiological simulations of influenza transmission on SES-heterogeneous networks.
(DOCX)

**S1 Table. ERGM model summary.** Summary of ERGM model results, including each model factor, its coefficient estimate and standard deviation, its p-value and a brief interpretation.
(DOCX)

**S2 Table. Variance inflation factors for ERGM covariates.** Higher values indicate greater correlation. VIF>20 is concerning. VIF >100 indicates severe multicollinearity.
(DOCX)

**S3 Table. Details of ERGM networks with added low SES nodes.**
(DOCX)

**S4 Table. Parameters for influenza network simulations.** Parameters are pertinent to influenza and to SES-based mechanisms. Parameters are defined as "high SES" and "low SES", though some of the "high SES" parameters are found from literature describing the entire population, due to lack of value specifically pertaining to those of high SES.
(DOCX)

**S5 Table. Covariate data for Bayesian hierarchical model.**
(DOCX)

## Acknowledgments

We thank Håvard Rue for his development of and assistance with the R-INLA package.

## Author Contributions

**Conceptualization:** Casey M. Zipfel, Vittoria Colizza, Shweta Bansal.

**Data curation:** Casey M. Zipfel.

**Formal analysis:** Casey M. Zipfel.

**Funding acquisition:** Shweta Bansal.

**Investigation:** Casey M. Zipfel, Shweta Bansal.

**Methodology:** Casey M. Zipfel, Vittoria Colizza, Shweta Bansal.

**Supervision:** Shweta Bansal.

**Validation:** Casey M. Zipfel.

**Visualization:** Casey M. Zipfel.

**Writing – original draft:** Casey M. Zipfel, Shweta Bansal.

**Writing – review & editing:** Casey M. Zipfel, Vittoria Colizza, Shweta Bansal.

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
