## [Decision Letter · Decision Letter 0]

16 Oct 2020

Dear Ms. Zipfel,

Thank you very much for submitting your manuscript "Health inequities in influenza transmission and surveillance" for consideration at PLOS Computational Biology. As with all papers reviewed by the journal, your manuscript was reviewed by members of the editorial board and by several independent reviewers. The reviewers appreciated the attention to an important topic. Based on the reviews, we are likely to accept this manuscript for publication, providing that you modify the manuscript according to the review recommendations.

Sincerely,

Alex Perkins

Associate Editor

PLOS Computational Biology

Virginia Pitzer

Deputy Editor

PLOS Computational Biology

[LINK]

Reviewer's Responses to Questions

**Comments to the Authors:**

Reviewer #1: Authors in this manuscript characterized the effects of health inequalities on an infectious disease dynamics. They developed a transmission model of influenza and assessed the role of SES-based behavioral and physiological differences on the disease dynamics at the population level.

A network model was developed using ERGM from the POLYMOD social network survey and integrated into the transmission model. The contact network structure accounts for heterogeneity in contact patterns by SES.

Five drivers of disparities in influenza burden that were accounted for in this study were: different social contact patterns, low vaccine uptake, low healthcare utilization, susceptibility, and low sickness absenteeism from school or work.

Further, the authors developed a spatial Bayesian hierarchical model to estimate latent influenza burden in low-SES populations in the United States.

This is a very important topic to focus on, considering the existing health disparities in the country, and this work addresses the association of SES levels with a disproportionate burden of disease and emphasizes the need to focus public health efforts on reducing socioeconomic health disparities.

Please see below my comments:

Looking at the network model goodness of fit figures, such as Figures S8 and S14, the simulated network is not in agreement with the POLYMOD data. Please discuss these results in the text.

Was the SEIR model calibrated before accounting for the five drivers?

In the equation for the probability of detection, page 19, please explain what z represents.

In the spatial model, the model is underestimating the outcome in comparison with the observed outcome. Please address this in the text. It also would be nice to see a time series of modeled vs. observed outcomes rather than scatter plots (Figs 34-37).

Page 20, lines 423-424, the covariate values were assumed to be constant over time from 2002-2008. Is it possible to develop the model for each year separately? Or develop a model with low, medium, and high values of covariates over the 2002-2008 time period? How would this change the results?

Page 21, lines 460-461: the missing covariate values were imputed. Could you please provide the percentage of data that was missing?

Reviewer #2: aba1238

Summary

This is an ambitious paper on a very important topic. It uses a mix of dynamical modeling and ecological statistical analysis to highlight the multiplicative effects of low SES on both flu burden and our persistent inability to observe the burden accurately. Especially at this moment where the COVID19 epidemic has laid bare the costs of systemic health inequity, this work using epi modeling and historical knowledge from flu adds useful evidence to advocate for more representative disease surveillance and more focused disease control.

The use of "intersectionality" is appropriate, and I appreciate bringing the language of social justice into model-heavy epidemiology, where it both belongs and was (to me) jarring at first. But it also motivates a question that readers will ask throughout -- are any inferences in this work likely to causal or even identifiable? Systemic inequity manifests in how many plausibly causal factors collide in the same population, and how non-causal factors often explain the most variance.

The dynamical-model-based exploration of how social factors coincide to drive transmission builds a clear and plausible case for how the factors pile together in the same direction, with reasonable numerical effect sizes. This speaks nicely to the compounding effects that can be quantified while individual causal factors cannot be described independently (as in the sense of a regression coefficent, all else held equal). This is the strongest part of the paper.

My main criticisms are in relation to that, in relationship to the ecological analysis that veers into causal over-interpretation in some places I flag below. Where noted, I think the storytelling subtracts more than it adds and so I encourage the authors to more carefully describe what can be learned from their analysis and what remains unidentifiable. To their point about intersectionality and the need for more effort to address the issues raised, not being able to answer the questions today because of fundamental statistical issues is one manifestation of the inequity they are shining the light on.

The open sharing in github is very welcome. I continue to be happy to see teams supporting this mode of science communication. I haven't vetted the code carefully, but I skimmed through some key functions and feel comfortable saying it is readable with a reasonable flow -- good for reproducibility and likely can be understood by a motivated reader. I appreciate the verbose variable names in the SEIR code in particular, I recognize the irritation of getting INLA outputs into a useable form, and must however acknowledge that the network code is noticeably less easy to make sense of.

I apologize that the review contains some comments I may be able to answer for myself given time -- this is a technically sophisticated effort that requires close attention to evaluate. COVID never sleeps but I must...

Overall I think this is an important paper that speaks to very pressing issues, and the methods are appropriate when interpreted judiciously. I look forward to seeing it in print after revision.

Major comments

37-39: It is correct to point out that all the factors are synergistic, but I'm not sure it's right to say that addressing one may alleviate others. Because one can't identify the relative contributions of each cause, it is possible to target the least important one and thus have very little benefit for the effort. Arguably this is the norm when targeting inequity in the US. I suggest going with a less causal statement.

When starting the results discussion around "low SES ILI", please be more clear about the meaning of terms. Lines 188 & 194 for example drop parenthetical definitions that are hard to keep track of in total. I take away that you define SES in terms of education only, and then look at ratios of ILI per hospital visit vs SES, but I'm not sure on first reading. This would also help (at least me) with my confusion about how the regression works (see my comment about line 412). I don't fully understand the outcome variable (it should be a count per the equation in line 393 but it appears to be a rate ILI/visits/1000??). The implementation in the code makes reference to an offset (https://github.com/bansallab/fluSES/blob/2624f6ade4230f94f1485a7590a69a10a5469a1a/Statistical%20Model/best_low_ses_county_inla_model_7_1.R#L187) so I assume the ratio is being done in a sensible way, but I don't have time to download and test myself nor should that be necessary.

Figure suggestion: can you show the low SES ILI incidence ratio vs the low SES variable? Or some other scatter plot that shows how visits fall off with SES and percent ILI rises. Something to emphasize the competing effects and clearly define the derived variable that is the outcome in the regression. This could strengthen the narrative about data inequity itself masking burden.

230-241: Here is where the causal storytelling I was worried about up front takes place! For example, household size is a strong predictor of transmission risk for COVID and also should be from a network perspective. So for household size to be be negatively associated with ILI risk, either the hypothesized social determinants are either stronger than the physical ones, or it is confounded with some unknown covariate (like how clusters with similar SES but different ethnic or religious backgrounds may have different family structures, and this covaries with geography too). Similarly, for flu vax, it's not unusual to find flu vax to be positively correlated with flu incidence. This can easily reflect general health-seeking behavior and not just vax-seeking due to risk, and thus there could be selection effects that go beyond what is scaled for with the total visits. The overall positive association with healthcare utilization factors points in this direction. My point is this paper has a strong message about colinear synergistic effects clearly aligning to enhance burden on low SES people, and low SES minimizes our system's ability to see that risk in data. That message is well told and in my opinion it is harmed by further adding on unsupported and incomplete causal scenarios.

254-267: great paragraph.

412: I'm confused about how the response can be normalized in a regression where the response variable is supposed to be a count. I'm not sure where my misunderstanding is arising, so please edit for clarity. Is it just that the variable is defined such that it's always positive and INLA handles the analytic continuation to non-integer counts gracefully?

Minor comments

Minor copy-editing required throughout. Like line 12, period-space before "Here".

Figure 2B needs a color legend on the figure itself.

416 and 423: "Inla does not allow..." in N-mixture models. Unless I'm misunderstanding what is meant by 'measurement covariates", this is not a general statement for all INLA models.

I did not evaluate the dynamical model code, but the method as described makes sense and the many supplemental figures document convincingly (to me) that the model is likely behaving as intended.

Reviewer #3: The manuscript analyzes the impact of socio-economic disparities in the spread of seasonal influenza in US. Authors combine mechanistic modeling of influenza spread on a contact network with statistical analysis of influenza incidence records across space. This allows them to quantify the relative role of different mechanisms determining increased spreading risk for low socioeconomic status (SES) individuals and to map health disparities in space.

The topic is an important and timely one. The manuscript presents an extensive analysis that combines different data sources and methodologies. I believe that the work has the potential to provide a nice contribution to PLOS Computational Biology. However, several improvements are needed especially in the presentation of the work that is at this stage unclear in many parts. Also, some hypotheses should be discussed more in depth, and alternative parameters should be explored in a sensitivity analysis. I detail in the following the major points to be addressed

1) The work is extensive and methods used are rich and complex. I believe that the methodological part should be put in prominence and should be presented before the results. Also, I could not follow the presentation of the results without reading the methods first.

2) More in general I believe that the paper should be restructured. Some parts of the methods are discussed in Results (e.g. end of page 10 when model validation is discussed). The Results section contains also some parts of the model discussion and limitations that should go on the Discussion (e.g. end of page 12, regarding the discussion of the vaccination result).

3) Many details are missing from the methods:

a. Authors mentions that the ERGM model is fitted to POLYMOD data. What observable is fitted?

b. In the description of the SES-based epidemiological model author write that delta and delta_low are respectively vaccination coverage in high and low SES individuals, however in TableS4 is written “general vaccination rate” for delta. This “general is confusing”, it points to an average quantity over the whole population. Similarly, for beta, gamma and roh it is not clear if these quantities are averages over the whole population or only high-income individuals

c. Table 4S should be presented in the main text.

d. How are the spreading simulations performed? More precisely: how are initial conditions defined? How long is the epidemic period (single season/multiple seasons)? Are people staying at home when infectious and how is this modelled in practice?

e. I had some difficulty in following the description of the Bayesian hierarchical model. Process predictor variables and covariates should be introduced immediately after the equations (at least briefly). I felt like plenty of details are given without a clear introduction of the overall methodology. Also, the way in which the two parts (network model and statistical analysis) are combined should be better presented.

4) Some assumptions should be discussed more in detail. Here are some assumptions/choice that I believe could be better motivated or would be benefit from sensitivity analysis.

a. Some sensitivity analysis on the parameters reported in Table S4 should be conducted. In particular I am referring to the ones related to the differences in transmission between high and low SES individuals.

b. Sensitivity should be conducted also on modelling assumptions. In particular, if I correctly understood authors assume that the whole population is naïve to the virus. In the modelling framework used by the authors, pre-existing immunity can be absorbed on the transmissibility parameter beta. However, some level of heterogeneity may in principle exist on the level of immunity among different SES groups. Authors should discuss this point, and test alternative scenarios.

c. Authors state “we resampled additional low education egos from the low education sample in the POLYMOD dataset”. This is not completely clear to me. Did authors test different proportion of low SES individuals in the population? What is the reasoning behind that?

**Have all data underlying the figures and results presented in the manuscript been provided?**

Reviewer #1: Yes

Reviewer #2: Yes

Reviewer #3: None

PLOS authors have the option to publish the peer review history of their article (what does this mean?). If published, this will include your full peer review and any attached files.

Reviewer #1: No

Reviewer #2: No

Reviewer #3: No
---

## [Editor Report · Decision Letter 1]

18 Dec 2020

Dear Ms. Zipfel,

We are pleased to inform you that your manuscript 'Health inequities in influenza transmission and surveillance' has been provisionally accepted for publication in PLOS Computational Biology.

Best regards,

Alex Perkins

Associate Editor

PLOS Computational Biology

Virginia Pitzer

Deputy Editor

PLOS Computational Biology

---

## [Editor Report · Acceptance letter]

15 Feb 2021

PCOMPBIOL-D-20-01395R1 

Health inequities in influenza transmission and surveillance

Dear Dr Zipfel,

I am pleased to inform you that your manuscript has been formally accepted for publication in PLOS Computational Biology. Your manuscript is now with our production department and you will be notified of the publication date in due course.

With kind regards,

Alice Ellingham
